# Athletes’ Opinions on Food Provision at European Athletics Championships: Implications for the Future

**DOI:** 10.3390/nu15020413

**Published:** 2023-01-13

**Authors:** Inês Maldonado, Catarina B. Oliveira, Pedro A. Branco, Mónica Sousa

**Affiliations:** 1Nutrition and Lifestyle, NOVA Medical School, Faculdade de Ciências Médicas, NMS, FCM, Universidade NOVA de Lisboa, 1169-056 Lisboa, Portugal; 2Comprehensive Health Research Centre (CHRC), NOVA Medical School, Faculdade de Ciências Médicas, NMS, FCM, Universidade NOVA de Lisboa, 1150-082 Lisboa, Portugal; 3Medical & Anti-doping Commission, European Athletics, 1003 Lausanne, Switzerland; 4CINTESIS@RISE, NOVA Medical School (NMS), Faculdade de Ciências Médicas, FCM, Universidade Nova de Lisboa, 1169-056 Lisboa, Portugal

**Keywords:** catering, competition events, dietary planning, nutrition labels, sport, sports nutrition

## Abstract

In competitive events, athletes’ performances can be affected by their food choices. In addition, nutrition labels are essential to sustain informed decisions and to allow athletes to comply with their dietary planning. Knowing what influences athletes’ food choices will help to improve the food provision in future championships. Therefore, we aimed to study the factors influencing athletes’ choices, their knowledge on nutrition labels, and their opinion on the food service at two European Athletics Championships. Questionnaires were completed by 339 athletes (57% males, 19.6 ± 1.3 years) competing at the 2019 European Athletics Under 20 and Under 23 Championships. Factors that may impact performance (time of the day and nutrient composition) were rated as important and very important by a higher percentage of athletes (78% and 74%, respectively) compared to the presence of teammates (32%) and the coach (23%). Among the athletes who knew what nutrition labels are (49%), 72% would like to have additional nutritional information in future championships. Furthermore, our study revealed that for most athletes (72%), food temperature is important or very important for food choices. Overall, food provision had positive results, but further research is needed to help organizers better tailor food provision to athletes’ needs.

## 1. Introduction

The importance of providing nutritionally adequate meals that optimize athletes’ performance is widely recognized [1]. During international competitions in particular, it is fundamental for athletes and coaches to feel confident with athletes’ food choices, making food provision a challenging responsibility.

Food choices affect athletes’ health and performance [2]; nevertheless, many factors may influence their preferences. A recent review [3] highlighted the ongoing interest in the factors that influence food choices and their predictors. According to it [3], individual, interpersonal, environmental, and policy factors may influence individuals’ food choices. For athletes, these characteristics may have an additional potential impact on sports performance, hence being critical [4]. However, few studies have explored what drives food selection in a competitive sports environment [2,5].

The food and drink service is one of the factors that has more value for the perception of overall quality in sports contexts [6,7]. Registered dietitians/nutritionists alongside with competition organizers have been working to improve the meal service at major international sports events [8,9,10,11,12,13]. In those competitions, catering must suit hundreds of athletes who come from very distinct cultures and have different preferences and needs, frequently with specific dietary requirements. These aspects make food provision a challenging task for caterers, who face additional constrains of time, temporary facilities, and casual staff [9]. Therefore, it is not unusual that meals do not conform to sports and general nutrition principles [12,14]. Additionally, the absence of international guidelines regarding catering for athletes allows different organizations to have distinct approaches [12].

European Athletics has developed nutritional guidelines that provide the local organizing committee, and the teams’ restaurants, guidance for food provision at European Athletics’ championships. Over the years, nutrition experts have improved and updated these guidelines to incorporate advances in sports nutrition research. Additionally, athletes’ opinions, collected in a systematic way, are an important asset for new and more directed approaches. One of the seven principles of quality management is an ongoing focus on improvement [15], therefore, other approaches regarding food provision should be considered. Nutrition labels have proven to be effective in assisting athletes with their food choices during international competitions. Additionally, the displayed information helps to reduce waiting time at dining areas [9,16]. In fact, providing the nutritional information of the food served is now mandatory at the Olympic Games and the Commonwealth Games [12].

Within athletics, elite athletes share common goals, particularly in regards to achieving maximum performance and protecting their health [17]. Nevertheless, it should be acknowledged that each athletic discipline has specific characteristics [17], and every athlete has his/her own unique issues regarding health, physique, performance, and preferences [1]. To the best of our knowledge, the factors influencing athletes’ food choice and opinion about the inclusion of nutrition labels at European Athletics Championships have not yet been assessed. Therefore, we aimed to study the athletes’ perspectives on the food service at the 2019 European Athletics Under 20 and Under 23 Championships, namely understanding what influences their food choices during competition, and their perceived relevance on having nutritional information available for future championships. Both the European Athletics Under 20 and the Under 23 Championships include 22 events of running, walking, jumps, throws, decathlon, and heptathlon, for both men and women [18,19]. Therefore, given the heterogeneity of events at these championships, and the specific nutritional strategies that best suit each athletic discipline, it is relevant to understand the preferences and needs of these athletes.

Therefore, we hypothesized that factors with a potential impact on performance could be important for athletes’ food choices and that nutrition labels could be helpful to sustain their decisions and to allow athletes to comply with their dietary planning.

## 2. Materials and Methods

### 2.1. Type of Study

This observational cross-sectional study was conducted during the 2019 European Athletics Under 23 (from 11–14 July, Gävle, Sweden) and Under 20 (from 18–21 July, Borås, Sweden) Championships.

The study protocol was approved by the Ethical Committee of NOVA Medical School|Faculdade de Ciências Médicas of the Universidade NOVA de Lisboa (57/2019/CEFCM).

### 2.2. Participants and Sample Size

All athletes over 18 years old competing at the 2019 European Athletics Under 23 or Under 20 Championships were invited to participate in the study. Out of the 2209 athletes competing at the 2019 European Athletics Under 20 (*n* = 1104) and Under 23 (*n* = 1105) Championships, 16% (*n* = 351) participated in our study; from these, 3% (*n* = 12) were excluded due to incomplete fulfilment of the questionnaire. Thus, the final sample comprised 339 athletes.

### 2.3. Instruments

A paper-pencil self-administered questionnaire, available as a Appendix A, was developed by the research team, based on other studies [8,11,16,20,21]. Participants took around 15 min to complete the questionnaire, which was written in English. Adequate English comprehension was assumed based on interviewer dialogue with the athlete during recruitment. The questionnaire included both closed- and open-ended questions. For most questions, five-point Likert-type scales were used. Factors influencing athletes’ food choices (smell, temperature, visual appearance, familiar food, nutrient composition, cooking method, stage of the competition, time of the day, presence of the coach, presence of teammates, and proximity to the stadium) were evaluated from 1 (not important) to 5 (very important). The frequency of utilization of nutrition labels was evaluated from 1 (never) to 5 (always). The components of a nutrition label (ingredient list, allergens, serving size, energy, carbohydrates, sugar, protein, fat, saturated fat, sodium/salt, icon representing a special regimen, icon representing an allergen, and icon representing the nutritional content of the dish) were evaluated from 1 (not important) to 5 (very important). The frequency of the ability to find adequate meals to supply dietary needs was evaluated from 1 (never) to 5 (always). The identification of different meal options was evaluated from 1 (never) to 5 (always). Additionally, the overall experience of eating at the team’s restaurant was evaluated from 1 (very bad) to 10 (excellent). Open-ended questions were included to allow comments, suggestions, and requests.

### 2.4. Procedure

Data collection took place during mealtime, at the teams’ restaurants. During these times, the research group approached the athletes and presented the aim and study protocol, as well as further information and explanations that they might have considered relevant. Participants signed an informed consent form and were invited to complete a self-administrated questionnaire. The completion of the questionnaire was performed in that moment, in the presence of the researchers. Additionally, participants could, at any moment, refuse to participate in the study without any kind of penalties involved.

### 2.5. Statistical Analysis

Data were entered and analyzed using the Statistical Package for Social Sciences (SPSS^®^) Statistical Software (version 26) [22]. To prevent manual data entry errors, the process was performed and checked by two researchers.

Athletes were grouped into six categories, similar to what was proposed by other authors [17], and according to their athletic discipline: sprints, middle-distance, distance, jumps, throws, and combined events.

For this observational study, data from both competition events were pooled and analyzed together. The normality of the two continuous variables (age and overall experience eating at the teams’ restaurants) was checked using the kurtosis and skewness coefficients [23]; since the coefficients were inferior to |2|, normality was assumed, and the descriptive statistics were presented using means and variances. Nominal or ordinal variables were described using frequencies. The Mann–Whitney test and Kruskal–Wallis test were used to compare nonparametric answers (i.e., differences between the answers of male and female athletes and between athletes from different athletic disciplines regarding the importance of nutrition label components). Post hoc tests were also performed [24]. The Student’s t-test and the one-way analysis of variance (ANOVA) for independent samples was used to compare parametric answers (i.e., differences on overall experience between male and female athletes and between participants from different athletic disciplines). The chi-squared test was used to analyze the significance of differences between categorical variables (i.e., the relationship between sex and knowing what a nutrition label is). Spearman’s correlation coefficient was performed to describe the association between two ordinal variables (i.e., the nutrition label utilization and the influence of nutrient composition on food choice). Statistical significance was set for *p* < 0.05.

Responses to open-ended questions were grouped according to topics for further analysis. The answers from the open-ended questions were coded, group into categories, and new variables were created for every main subject of answers; moreover, we selected a representative example for each category.

## 3. Results

### 3.1. Participant Characteristics

The final sample comprised 339 athletes aged between 18 and 22 years old (19.6 ± 1.3 y), representing 35 countries. Participant characteristics are displayed in Table 1. Most participants (82%) reported having past international competition experience.

Thirteen percent of the respondents reported a specific food restriction. Particularly, 21 athletes reported having an allergy/avoidance (dairy or lactose, fish, fruits, gluten, nuts, seafood, or certain vegetables), 6 reported being vegetarian, 8 reported avoiding red meat, 2 reported having food restrictions related to religious beliefs (such as pork), and 7 athletes reported other restrictions (i.e., avoiding fried foods).

### 3.2. Factors Influencing Food Choice during Competition

The time of the day and the nutrient composition were the factors that athletes reported to be more important, with 78 and 74% of athletes, respectively, rating them as important or very important. In contrast, the presence of the coach and the presence of teammates were considered less relevant, with 53% and 43% of athletes, respectively, rating them as not important or less important (Figure 1). Competition-related factors were considered relevant for athletes’ food choices, with the stage of competition and the proximity to the stadium being rated as important or very important by 69% and 63% of athletes, respectively. Sprinters were more likely to rate the visual appearance as important or very important, compared to throwers (77% vs. 50%; *p* = 0.03) and jumpers (77% vs. 49%; *p* = 0.015). Distance runners were more likely to rate the stage of competition as important or very important, compared to throwers (81% vs. 62%; *p* < 0.001). Athletes who had past international competition experience were more likely to consider the nutrient composition (78% vs. 51%; *p* < 0.001) and the cooking method (49% vs. 33%; *p* = 0.007) important or very important, compared to those who did not have this experience. There was no relationship between sex and any factor influencing food choice during the competition (*p* > 0.05).

### 3.3. Nutrition Labeling

Almost half of the athletes (49%) reported knowing what a nutrition label is. Among those, nutrition labels were always (11%), often (36%), sometimes (34%), occasionally (16%), or never (4%) used. A higher frequency of nutrition label utilization was positively associated with higher rates on the influence of nutrient composition for food choice (ρ = 0.186; *p* = 0.018). There was no relationship between sex and knowing what a nutrition label is (*p* = 0.653) and the frequency of nutrition label utilization (*p* = 0.082). Similarly, there was no relationship between having had past international competition experiences and knowing what a nutrition label is (*p* = 0.189) and the frequency of nutrition label utilization (*p* = 0.214). Athletes from different athletic disciplines reported different knowledge regarding nutrition labels (*p* < 0.001), with most distance (74%) and middle-distance runners (67%) reporting knowing what a nutrition label is, contrasting with combined events athletes (50%), jumpers (35%), and sprinters (35%).

Most athletes (72%) would like to have the nutritional information of the meals available in future championships, and 73% considered that this information would help them to make better food choices. Additionally, wanting nutrition labels in future competitions was positively associated (*p* < 0.001) with considering that nutrition labels help in making better food choices. Additionally, when athletes were asked to describe why nutrition labels would be helpful, the most referred to advantages were the assistance in adequate food choice and the knowledge about the ingredients of each meal (Table 2).

Concerning nutrition label components (Figure 2), protein (89%) and carbohydrates (89%) were the categories being rated as important or very important by a greater number of athletes, followed by sugar (84%) and energy (83%). On the other hand, serving size was the component which fewer athletes (58%) rated as important or very important. However, athletes who have had past international competition experiences were more likely to consider the serving size as important or very important, compared to those who have not had the experience (62% vs. 32%; *p* = 0.021). Middle-distance runners were more likely to consider the presence of the protein content in a nutrition label as important or very important when compared to sprinters (98% vs. 80%; *p* = 0.03). There was no relationship between sex and the opinion on how important the presence of any component of a nutrition label was (*p* > 0.05).

### 3.4. Overall Opinion

The ability to find adequate meals to supply dietary needs was reported as always (18%), often (46%), sometimes (29%), occasionally (6%), or never (1%) able to find those options. Results were independent of having had previous international competition experience (*p* = 0.068), sex (*p* = 0.143), and the athletic discipline (*p* = 0.065). When asked what would help them to locate a particular food item, 36% of the athletes responded that the existence of nutrition labels would facilitate this identification. Other suggestions are displayed in Table 3.

Finally, the overall eating experience at the teams’ restaurants was 7.5 ± 1.6, on a scale of 1 to 10. There was no relationship between the reported overall experience and sex (*p* = 0.056), having had previous international competitions experience (*p* = 0.520), nor within participants from different athletic disciplines (*p* = 0.062).

## 4. Discussion

The current study gathered a singular and unique perspective of European athletes on factors influencing their food choices under a high-level and high-pressure competition environment and on the provision of nutrition labels in future athletic championships.

Concerning food choice and similar to what was found in recent studies [4,5], factors that could impact sports performance (time of the day, nutrient composition, and cooking method) were considered more important than interpersonal factors (presence of coach and teammates). The latter, especially the influence of teammates, seems to determine the food choices of college athletes [25,26]; however, this influence loses importance as athletes’ experiences increase [26]. In parallel, enhancing athletic performance earns relevance as athletes’ experiences increase [2]. In consonance with this, in our study, cooking method and nutrient composition had higher importance for athletes who had previous international competition experience compared to less experienced athletes.

In our study, athletes gave greater importance to sensory properties (visual appearance, temperature, and smell) and less importance to the familiarity of the food. Conversely, other studies conducted at major international competitions [4,5] found that the familiarity of the food was one of the main factors influencing athletes’ food choices. A possible explanation for this might be that the 2019 European Athletics Under 20 and Under 23 Championships took place in Sweden, and our study only included European participants. In addition, the menus initially proposed by both local organizing committees were revised and adjusted by the experienced European Athletics sports nutritionist, who adjusts the menus according to the European Athletics Nutrition Guidelines. Nevertheless, sensory properties are decisive factors for food choice [12,27]. Compared to non-sensorial factors, and for the general population, taste, texture, odor, and appearance are important factors for food choice [27]. Interestingly, in our study, the temperature—a factor for which importance has not been highlighted in recent studies—was the third most important factor highlighted by the athletes. Therefore, we suggest that future studies include temperature as a variable of interest for athletes’ food choice, to better understand the relevance of this factor. Our results also highlight the necessary cooperation between caterers and specialized sports dietitians/nutritionists to develop meal plans that meet both athletes’ dietary needs and requirements, and also suitable organoleptic characteristics, as suggested in the past [8].

Regarding nutrition labels, most athletes would like to have this information in future events. Similar to a study with athletes from several sports competing at the 2010 Commonwealth Games [16], in our study athletes mentioned that nutrition labels would help them choose an adequate meal during a competition. The nutrient composition was rated as important by most athletes; thus, it was not surprising to see that among those who knew what a nutrition label is, nutrition labels were frequently used. Our data also suggest these athletes recognize the relative importance of energy and macronutrients, such as carbohydrates and protein, since these nutrition label components were among the most valued. Our results, namely the high value assigned to nutrient composition, the frequent utilization of nutrition labels, and the percentage of athletes who would like to have nutritional information in future championships, highlight that in upcoming European Athletics’ championships labeling the food options is highly relevant. This suggestion is aligned with what is now required by the Olympic and Commonwealth Games [12].

High-performance athletes share common objectives, yet the athletics includes numerous disciplines, and each one encompasses specific characteristics determining different nutritional needs [17]. This might explain the divergence of opinions found in our study. For example, distance runners, who are highly advised to include strategies to store muscle glycogen prior to the race [28], considered the stage of competition more important than throwers, who do not depend considerably on these nutritional strategies. Our data revealed that middle-distance runners considered protein to be a more important macronutrient compared with sprinters. This is an interesting result since, for many years, strength-power athletes, but not middle-distance runners, have been encouraged to have high-protein diets [29] and, in a previous study, power/sprint athletes were more likely to follow a high-protein diet compared to other athletes [21]. Nevertheless, both groups rely on the intake of carbohydrates and protein in the hours after the competition to recover between multiple races [17,29,30].

We did not find any differences between male and female athletes. Regarding factors influencing food choice, our data contrast with a previous study in which female athletes were more likely to rate the familiarity and smell of food as important, compared to male athletes [4]. Concerning nutrition labels, a previous study found that female athletes considered providing nutrition information more important than male athletes [16], which also differs from our results. Nevertheless, these authors did not find an association between sex and the use of nutrition labels [16], similar to our results.

A buffet-style service is considered the most appropriate way of serving athletes [31], and thus should be promoted. Additionally, it is recommended by the European Athletics guidelines that food organization in the buffet line follows the order of a traditional meal, presenting the salad buffet and the vegetable soup first, then the carbohydrate’s garnish, subsequently the fish, meat, and vegetarian dishes and, at the end of the buffet line, the desserts. As the search for food was considered easy for most athletes in our study, the food display seems to be adequate. Nevertheless, in open-ended questions, many athletes added that they would like to have nutrition labels, suggesting that this may be an area of improvement. To the date of these championships, the European Athletics recommended that the name of the dish should be displayed near each food item, including the cooking method. Our results suggest both (a) the need for additional supervision to help caterers in fulfilling this recommendation and (b) the improvement of the labels.

Registered dietitians/nutritionists have been collaborating with competition organizers to ensure that meals are nutritionally adequate for athletes [8,9]. Our positive results on the overall teams’ restaurant experiences reinforce the relevance of developing nutritional guidelines for the food supply during international championships. We believe that these results also reflect the importance of having nutritional guidelines created specifically for athletics and European Athletics’ championships. It also supports the importance of the dialogue between competition organizers and caterers [8,10,11]. Effective communication between the organizers and the catering company should exist, and all the aspects of the food provision must be discussed and agreed on before the competition starts. This advanced planning allows both organizers and caterers to adjust details and discuss possible necessary adaptations to the already established guidelines. Additionally, to guarantee that all the procedures and food options comply with those previously agreed upon and that any issue that arises during competition is handled in the best possible way (both for the organizer and the catering partner), the food provision should be monitored by a nutrition expert during the competition event.

Our study has some limitations that should be acknowledged, namely the fact that the questionnaire has not been validated. Additionally, there were language issues, which may have excluded some non-English-speaking athletes from participating in the championships. Although adequate English comprehension was assumed based on interviewer dialogue with the athlete at recruitment, it may not have reflected the athlete’s reading literacy and, as a result, respondent miscomprehension errors may have occurred. Nevertheless, researchers were present while athletes were completing the questionnaire to clarify any information if needed. In addition, participants were frequently sited next to peers and coaches, which may have influenced their responses. Finally, a convenience sample was recruited; consequently, it may not represent all athletes that competed at the two championships. Nevertheless, and although we were only able to include 339 participants from the 2209 athletes who participated at the 2019 European Athletics Under 20 (*n* = 1104) and Under 23 (*n* = 1105) Championships, we were able to collect answers from high-level athletes from 35 nationalities.

## 5. Conclusions

In conclusion, our study has gathered a singular and unique perspective of European athletes from athletics in a high-level and high-pressure competition environment. Factors related to athletic performance (time of the day and nutrient composition) proved to be particularly important for athletes’ food choices. In addition, the temperature of food was considered a relevant factor by the athletes. Overall, the athletes’ opinions about the food service at the 2019 European Athletics Under 20 and Under 23 Championships were highly positive. Nevertheless, athletes’ suggestions are essential to improve future catering services, namely the inclusion of nutrition labels in future championships. Altogether, our data support the importance of specific nutritional guidelines for catering during international sporting events to ensure uniformity and to increase athletes’ confidence in the food served. Furthermore, our data reinforce the importance of promoting food literacy and nutrition education among athletes, as half of the athletes did not know what a nutrition label is. Future research on factors influencing food choice and the provision of point-of-choice nutrition labels during international championships is warranted to aid caterers and organizers in tailoring food provision to the athletes’ needs and preferences.

## Figures and Tables

**Figure 1 nutrients-15-00413-f001:**
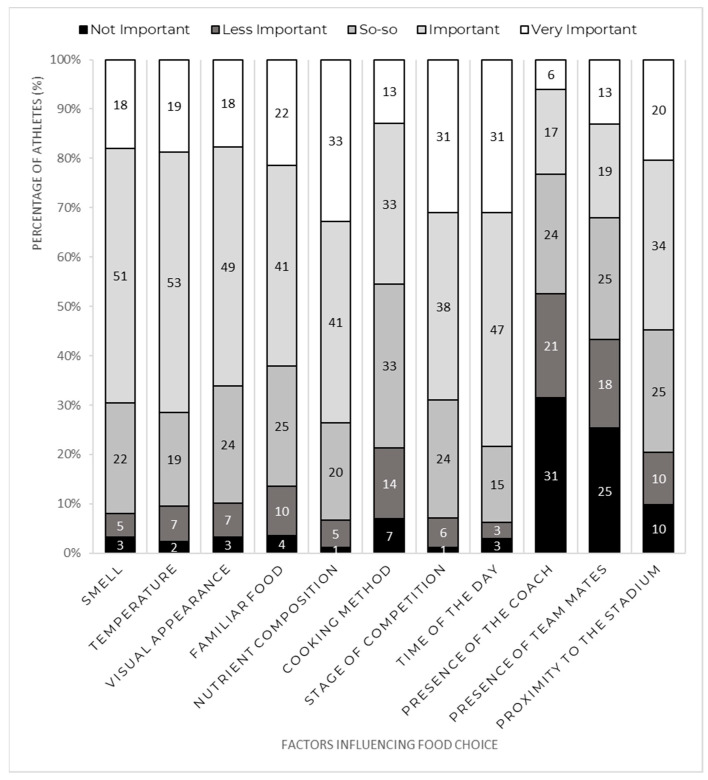
Factors influencing food choice (%).

**Figure 2 nutrients-15-00413-f002:**
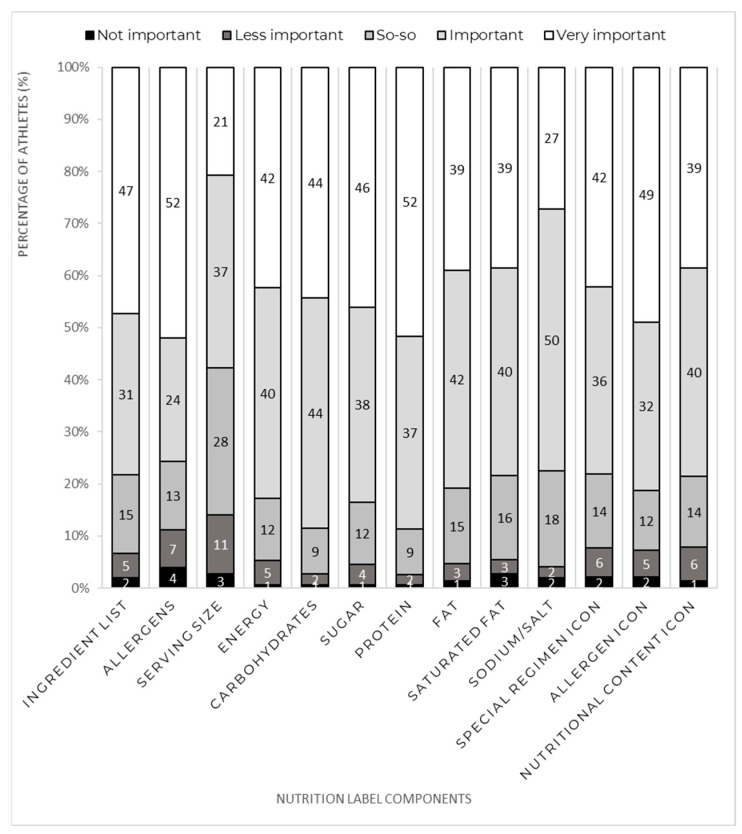
Relative importance of the nutrition label components (%).

**Table 1 nutrients-15-00413-t001:** Participant characteristics.

	*n* (%)
Female	145 (43%)
Male	194 (57%)
U20 ^a^	166 (49%)
U23 ^b^	173 (51%)
Never attended school	3 (1%)
Completed intermediate/middle school	48 (14%)
Completed high school	124 (37%)
Attended university or other tertiary institution	160 (48%)
Sprints ^c^	120 (35%)
Middle-distance ^d^	68 (20%)
Jumps ^e^	59 (17%)
Throws ^f^	54 (16%)
Distance ^g^	31 (9%)
Combined events ^h^	7 (2%)
Great Britain and Northern Ireland	43 (12.7%)
Spain	29 (8.6%)
Sweden	29 (8.6%)
Portugal	24 (7.1%)
Ireland	23 (6.8%)
Belgium	17 (5.0%)
Finland	16 (4.7%)
Italy	15 (4.4%)
Germany	12 (3.5%)
Greece	11 (3.2%)
Hungary	11 (3.2%)
Netherlands	11 (3.2%)
Austria	9 (2.7%)
Slovenia	9 (2.7%)
Ukraine	9 (2.7%)
Belarus	7 (2.1%)
Croatia	7 (2.1%)
France	7 (2.1%)
Poland	7 (2.1%)
Bulgaria	6 (1.8%)
Czech Republic	5 (1.5%)
Latvia	5 (1.5%)
Switzerland	5 (1.5%)
Estonia	4 (1.2%)
Bosnia and Herzegovina	3 (0.9%)
Serbia	3 (0.9%)
Iceland	2 (0.6%)
Norway	2 (0.6%)
Romania	2 (0.6%)
Azerbaijan	1 (0.3%)
Cyprus	1 (0.3%)
Denmark	1 (0.3%)
Liechtenstein	1 (0.3%)
Lithuania	1 (0.3%)
Luxembourg	1 (0.3%)

a—European Athletics Under 20 Championship; b—European Athletics Under 23 Championship; c—100, 200, 400, 100, or 110 m hurdles, 400 m hurdles, 4 × 100 m, 4 × 400 m; d—800 m, 1500, 3000, 5000, 3000 m steeplechase; e—high jump, pole vault, long jump, triple jump; f—shot put, discus throw, hammer throw, javelin throw; g—10,000 m, 10 or 20 km walk; h—heptathlon, decathlon.

**Table 2 nutrients-15-00413-t002:** Reasons regarding the usefulness of nutrition labels.

Response Category	Athletes (%)	Example of Responses
Adequate choice	25	To choose healthier meals
Knowing the ingredients	23	It helps me to know the components of the dish
Not useful	18	Because I will just eat what I normally eat at home
Nutritional composition	17	Because athletes would know what each meal has in terms of nutrients
Improving athletic performance	8	Because I think it is important for a better performance
Other answers	8	Because it is a tool in which I trust

**Table 3 nutrients-15-00413-t003:** Athletes’ written suggestions to the question: “what do you think could help you find/locate a certain item (to supply your dietary needs)?”.

Response Category	Athletes (%)	Example of Responses
Labels	36	More labels to clarify the meaning of each of the food items
Signs, figures, or icons	30	Signs for the different foods’ location
Organization, meal display, or type of service	16	Different sectors with certain foods
Name, identification, or ingredient list	15	To have a written name for each of the food items
Others	11	To have access to other teams’ restaurants

## Data Availability

The data are not publicly available due to privacy and ethical restrictions.

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
