# Peer review of "Athletes’ Opinions on Food Provision at European Athletics Championships: Implications for the Future"

_nutrients, 2023, doi:10.3390/nu15020413_

Round 1

Reviewer 1 Report

Your paper is straight forward, well-written and quite interesting. Observations such as the ones made in this paper are important to athletes, coaching and athletic governing bodies.

I just have a few suggestions to improve the manuscript:

Abstract:

Not clear what you mean here: championships. “Furthermore, our study revealed that for most athletes (72%) temperature is important or very important for food choices.” Do you mean: food temp, body temp, air temp, their mood state?

Line 42: suit not suite

Line 43: delete comma

Line 49: Provide a reference to the guidelines

Line 77: delete comma

Line 78-80: What do you mean by district opinions? I don’t think you need those lines. It is not specific enough to be a hypothesis or secondary hypothesis.

Lines 158-167: Would be easier to read this in a table.

Line 263: delete comma

Reviewer 2 Report

L35: Not sure what is being conveyed here

“Concerning athletes, aspects with a potential impact on sports performance may be critical.” 1) I think you are suggesting that “things’ may impact sports performance.  In particular – nutrition.  2) Is there a better way to state this because as it is written “aspects” is not clear.

L38: lower case athletics, nutritional, guidelines, local, organizing, committee, athletics

Is European Athletics a proper name?

L64: cross-sectional

Table 1- is it necessary to have the title repeated inside the table?

L130 – better word than “ones”:  ….were the questions …were the categories

L183: I think it could read better revised:

The ability to find adequate meals to supply dietary needs was reported as: always (18%)….

L193: studies not researches
